# Epidemiology of infection by pulmonary non-tuberculous mycobacteria in French Guiana 2008–2018

**Milène Chaptal**[1,2]*, **Claire Andrejak**[3], **Timothée Bonifay**[4], **Emmanuel Beillard**[5], **Geneviève Guillot**[6], **Stéphanie Guyomard-Rabenirina**[7], **Magalie Demar**[8], **Sabine Trombert-Paolantoni**[9], **Veronique Jacomo**[10], **Emilie Mosnier**[1], **Nicolas Veziris**[11], **Felix Djossou**[1], **Loïc Epelboin**[1,12], **French Guiana PNTM working group**[¶]

**1** Tropical and Infectious Diseases Department, Andrée Rosemon Hospital, Cayenne, French Guiana, **2** Pneumology Department, University Hospital of Guadeloupe, Pointe-à-Pitre, France, **3** Pneumology Department, University Hospital, Amiens, France, **4** Penitentiary ambulatory care and consultation unit, Andrée Rosemon Hospital, Cayenne, French Guiana, **5** Pasteur Institute of Guiana, Cayenne, French Guiana, **6** Medical Department, Andrée Rosemon Hospital, Cayenne, Guyane française, **7** Transmission, Reservoir and Diversity of Pathogens Unit, Pasteur Institute of Guadeloupe, Les Abymes, Guadeloupe, **8** Laboratory, Andrée Rosemon Hospital, Cayenne, French Guiana, **9** Cerba Europeanlab, St Ouen l'Aumone, France, **10** Biomnis Laboratory, Lyon, France, **11** Sorbonne Université, INSERM U1135, Centre d'Immunologie et des Maladies Infectieuses (CIMI-Paris), Centre National de Référence des Mycobactéries et de la Résistance des Mycobactéries aux Antituberculeux, Département de Bactériologie, Groupe hospitalier APHP, Sorbonne Université, Site Saint-Antoine, Paris, France, **12** Centre d'investigation Clinique INSERM 1424, Centre Hospitalier de Cayenne, Andrée Rosemon, Cayenne, French Guiana

¶ French Guiana PNTM working group. Membership of French Guiana PNTM working group is provided in the Acknowledgments
* mi.chaptal@yahoo.com

**Data Availability Statement:** All relevant data are within the manuscript.

**Funding:** The author(s) received no specific funding for this work.

## Abstract

### Introduction

Unlike diseases caused by *Mycobacterium tuberculosis*, *M. leprae* and *M. ulcerans*, the epidemiology of pulmonary non-tuberculous mycobacteria (PNTM) has not received due attention in French Guiana. The main objective of the current study was to define the incidence of these PNTM infections: NTM pulmonary diseases (NTM-PD) and casual PNTM isolation (responsible of latent infection or simple colonization). The secondary objectives were to determine species diversity and geographic distribution of these atypical mycobacteria.

### Methods

A retrospective observational study (2008–2018) of French Guiana patients with at least one PNTM positive respiratory sample in culture was conducted. Patients were then classified into two groups: casual PNTM isolation or pulmonary disease (NTM-PD), according to clinical, radiological and microbiological criteria defined by the American Thoracic Society / Infectious Disease Society of America (ATS / IDSA) in 2007.

### Results

178 patients were included, out of which 147 had casual PNTM isolation and 31 had NTM-PD. Estimated annual incidence rate of respiratory isolates was 6.17 / 100,000 inhabitants per year while that of NTM-PD was 1.07 / 100,000 inhabitants per year. Among the 178

**Competing interests:** The authors have declared that no competing interests exist.

patients, *M. avium complex (MAC)* was the most frequently isolated pathogen (38%), *followed by M. fortuitum* then *M. abscessus* (19% and 6% of cases respectively), the latter two mycobacteria being mainly found in the coastal center region. Concerning NTM-PD, two species were mainly involved: *MAC* (81%) and *M. abscessus* (16%).

## Discussion/Conclusion

This is the first study on the epidemiology of PNTM infections in French Guiana. PNTM's incidence looks similar to other contries and metropolitan France and NTM-PD is mostly due to *MAC* and *M.abscessus*. Although French Guiana is the French territory with the highest tuberculosis incidence, NTM should not be overlooked.

## Introduction

Mycobacteria are widespread acid-fast bacilli (AFB) belonging to the genus *Mycobacterium* which include over 190 species. Non-tuberculous mycobacteria (NTM) have been poorly studied, while contrastingly, *Mycobacterium tuberculosis* complex and *Mycobacterium leprae*, human causative agents of Tuberculosis and Leprosy, have been studied in great detail. NTM are ubiquitous bacteria found in soil and water sources. Human contamination usually occurs by inhalation or skin-penetration [1]. NTMs mostly affect lungs and also others organs such as skin and soft tissues, bones, the lymphatic system etc. Buruli ulcer, caused by *Mycobacterium ulcerans*, is well known in French Guiana, with the tissue destruction resulting in skin ulcers and in severe cases causing bone infection [2]. Respiratory strains, called pulmonary-NTM (PNTM), can colonize respiratory airways without invading pulmonary tissues and can cause either an indolent infection or progress to an invasive disease. The unfortunate main complication is the progression to chronic respiratory failure. Among PNTM infections, the diagnostic criteria of the American Thoracic Society / Infectious Disease Society of America (ATS / IDSA) in 2007, enabled to distinguish PNTM isolation, including colonization and indolent infection, from true NTM pulmonary disease (NTM-PD) [3]. Latest guidelines in 2020 kept the same criteria, supplementing it with the precision of an minimal interval between two samples [4].

French Guiana is a tropical French overseas territory. It is located on the northeast Atlantic coast of the South American continent, between Brazil and Suriname. There are marked environmental and socio-demographical differences with mainland France. This territory is scarcely populated (about 250,000 inhabitants for 83,500 km$^2$) and mostly covered by the Amazon forest [5]. French Guiana is the French region with the highest Tuberculosis incidence (32.5 per 100,000 inhabitants in 2017) [6]. Although the epidemiology of Tuberculosis, Leprosy and Buruli ulcer in French Guiana has been studied extensively, the literature on the epidemiology of NTM-PD is scarce [2,7–11].

Considering the scarcity of knowledge on PNTM infections in French Guiana, we conducted a retrospective and descriptive epidemiological study from 2008 to 2018 on patients living in the region. The main objective was to define the incidence of PNTM infections. Secondary objectives were to determine species diversity and geographic distribution of these atypical mycobacteria in French Guiana.

## Methods

### Ethics statement

This is a retrospective, non-interventional study and the data were anonymized for all patients. We only collected data necessary for the purpose of our study. The data collected will remain

in an archive for 15 years. This research was in compliance with the law "Informatique et Libertés" of January 6, 1978 as amended and the law No. 2018–493 of June 20, 2018 on the protection of personal data. Is also in compliance with European Parliament guidelines in April 27, 2016. Finally, data were transferred and collected in accordance with the reference methodology MR003 of the Commission Nationale de l'Informatique et des Libertés (CNIL) for which the Centre Hospitalier de Cayenne has signed a compliance commitment. The database was declared to the CNIL and approved on April 19, 2019 (declaration number: 2212828v0).

## Study design

We performed an observational and descriptive study on patients from the three French Guiana general hospitals: Cayenne, Kourou and Saint Laurent du Maroni.

## Inclusion and exclusion criteria and case definition

All patients living in French Guiana and having at least one NTM positive culture of a respiratory sample were included between January 1, 2008 and December 31, 2018. Patient identification was based on respiratory samples obtained from all laboratories analyzing mycobacteria in French Guiana, namely: Pasteur Institute of French Guiana, Pasteur Institute of Guadeloupe, and private laboratories in mainland France Cerba and Biomnis. All medical records were reviewed and patients were classified into 2 categories: patients with diagnostic criteria of the ATS / IDSA 2007 fulfilled and patients who did not fulfill all the criteria but had casual PNTM isolation. The criteria are as follows: 1) clinical and radiological criteria: presence of symptoms, imaging compatible with cavities, nodules, micronodules associated with bronchiectasis, and the exclusion of other diagnoses. 2) microbiological criteria: presence of at least two separate sputum or gastric tubing or one positive bronchoalveolar lavage in culture. Excluded cases were sampling errors (*M. tuberculosis* or isolated extra-pulmonary NTM) and patients who could not be categorized after collegial discussion between lung and infectious disease specialists of the Cayenne general hospital.

## Data collection and analysis

The data was collected and anonymized on Excel. For the analysis, we used the software Stata (version 12). Incidence assessment was based on the census of the National Institute of Statistics and Economic Studies (INSEE). For binary qualitative variables, $Chi^2$ or Fisher test were used. Finally, we used Mapinfo software (version 12) to make a distribution map of respiratory isolates, according to the geographical definition of the 4 regions in French Guiana: Coastal center, Savannas, West and East French Guiana.

## Results

265 patients were identified between January 1st, 2008 to December 31st, 2018. 178 patients had a positive respiratory sample in culture (Fig 1). Thirty-one NTM-PD patients (17%) met the ATS / IDSA diagnostic criteria. Demographic characteristics are summarized in Table 1.

Mean annual incidence over the study period was 6.17 / 100,000 inhabitants / year for overall PNTM positive respiratory sample and 1.07 cases / 100,000 inhabitants / year for NTM-PD. Incidence of NTM-PD remained stable over the years with a slight increase since 2017. The number of casual PNTM isolation tended to increase from 2013 onwards (Fig 2).

Culture positive samples were mainly obtained from sputum (64%). Bronchoalveolar lavage (BAL) was performed in 35% of NTM-PD (11 out of 31 patients). No pleural fluid, protected brushing, trans-bronchial or surgical lung biopsy were analyzed. The main strain identified

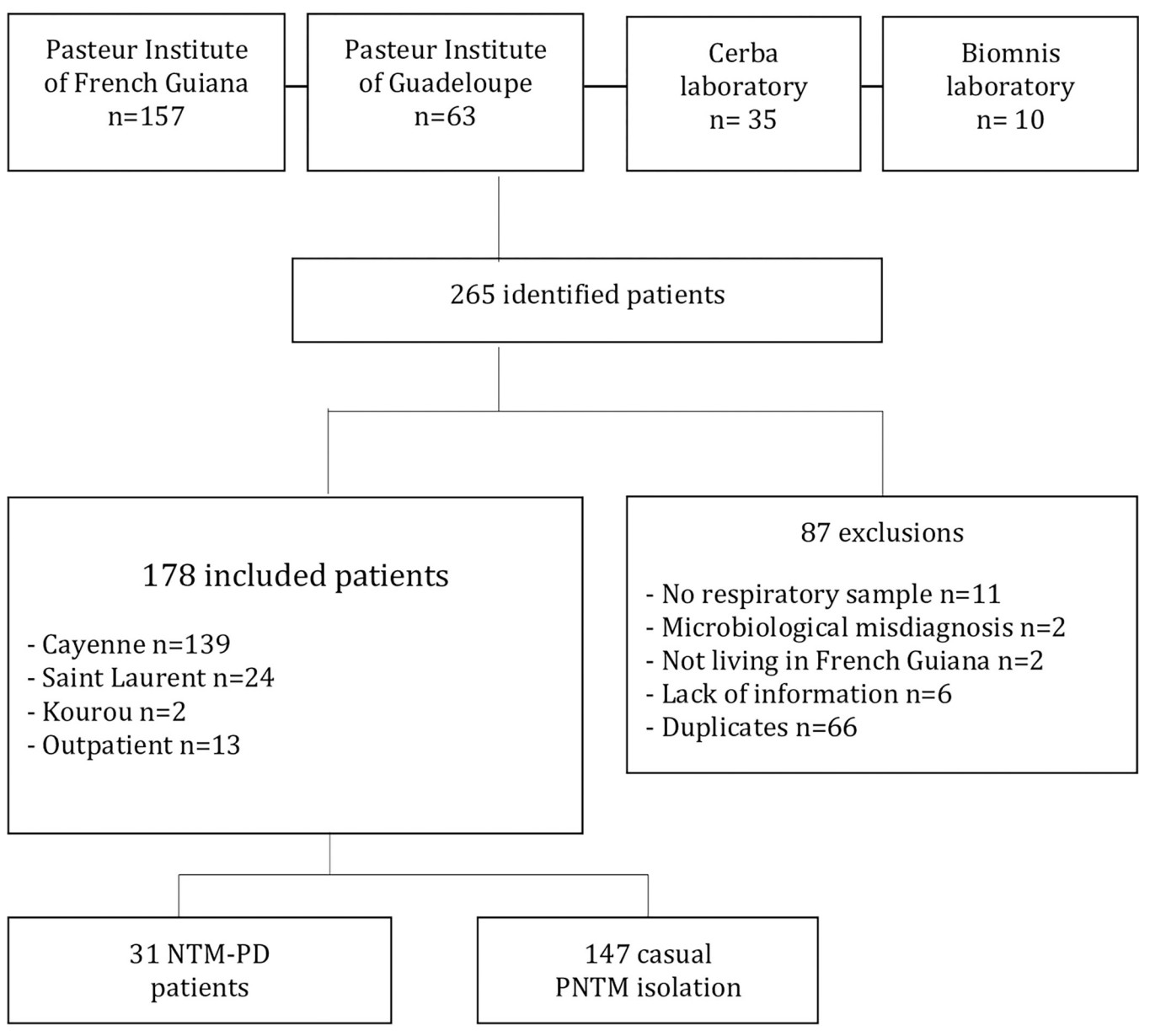

PNTM: Pulmonary Non-Tuberculous Mycobacteria

NTM-PD: Non-Tuberculous Mycobacteria Pulmonary Disease

**Fig 1. Study flow chart.**

was *Mycobacterium avium* complex (*MAC*), which was seen in 68 patients (38%), followed by *M. fortuitum* in 34 patients (19%) and *M. abscessus* group in 10 patients (6%) (Tables 2 and 3). *MAC* and *M. abscessus* were significantly associated with pulmonary disease ($p<0.001$ and $p = 0.02$ respectively). *M.fortuitum* and unidentified and non-pathogenic mycobacteria were significantly associated with a lack of disease ($p<0.001$, $p = 0.01$ and $p = 0.03$ respectively). 31

**Table 1. Demographic characteristics and comorbidities.**

| Caracteristics | Patients | |
|---|---|---|
| | N = 178 | % |
| **Demographic data** | | |
| Men | 108/178 | 61 |
| Children (< 18y) | 3/178 | 2 |
| Age (mean, y), (min–max) | 49 | 9–90 |
| Native country: | | |
| French Guiana | 34/155 | 22 |
| Neighboring countries (Brazil, Surinam, Guyana) | 66/155 | 43 |
| Haiti | 31/155 | 20 |
| Precarous living conditions * | 96/149 | 64 |
| No heath insurance coverage | 86/144 | 60 |
| **Comorbidities** | | |
| Chronic pulmonary disease | 57/174 | 33 |
| Chronic Obstructive Pulmonary Disease | 31/ 174 | 17 |
| Bronchiectasies | 8/ 174 | 5 |
| Tuberculosis history | 27/174 | 16 |
| Heart disease history | 35/174 | 20 |
| Arterial hypertension | 28/174 | 16 |
| Diabetes | 12/174 | 7 |
| Cerebrovascular disease** | 9/174 | 5 |
| HIV Infection | 77/167 | 46 |
| CD4 < 200 | 49/72 | 68 |
| CD4<50 | 28/72 | 39 |
| Immunosuppressive therapy | 7/178 | 4 |
| Solid tumors | 4/178 | 2 |
| Undernutrition (BMI<18,5) | 37/163 | 23 |
| Gastroesophageal reflux or gastritis | 49/175 | 28 |
| Smoking | 85/148 | 57 |
| CCI score *** | | |
| Low | 46/176 | 26 |
| Medium | 92/176 | 52 |
| High | 38/176 | 22 |

* *Homeless*, *Prisoners*, *Jobless*, *No health insurance coverage*, *without social subsidies*

** Cerebrovascular disease = 7 stroke, 1 cerebral amyloidosis et 1 mental retardation

*** CCI = Charlson Comorbidity Index simplify Low (score = 0)- medium (1–2)- High (3 or more) among: diabetes, chronic pulmonary disease, connective tissue disease, AIDS, moderate or severe kidney or liver failure, cerebrovascular disease, hemiplegia, dementia, solid tumors, hemopatologic maligniancies.

out of 147 casual PNTM isolation were unidentified species (Table 2). Geographical distribution of PNTM was heterogeneous, with a predominance of Rapid Growing Mycobacteria (RGM) in the coastal central region while *MAC* showed an even distribution across the region (Fig 3).

## Discussion

To the best of our knowledge, this is the first study describing the incidence and microbiological epidemiology of infections linked to NTM with respiratory expression in French Guiana.

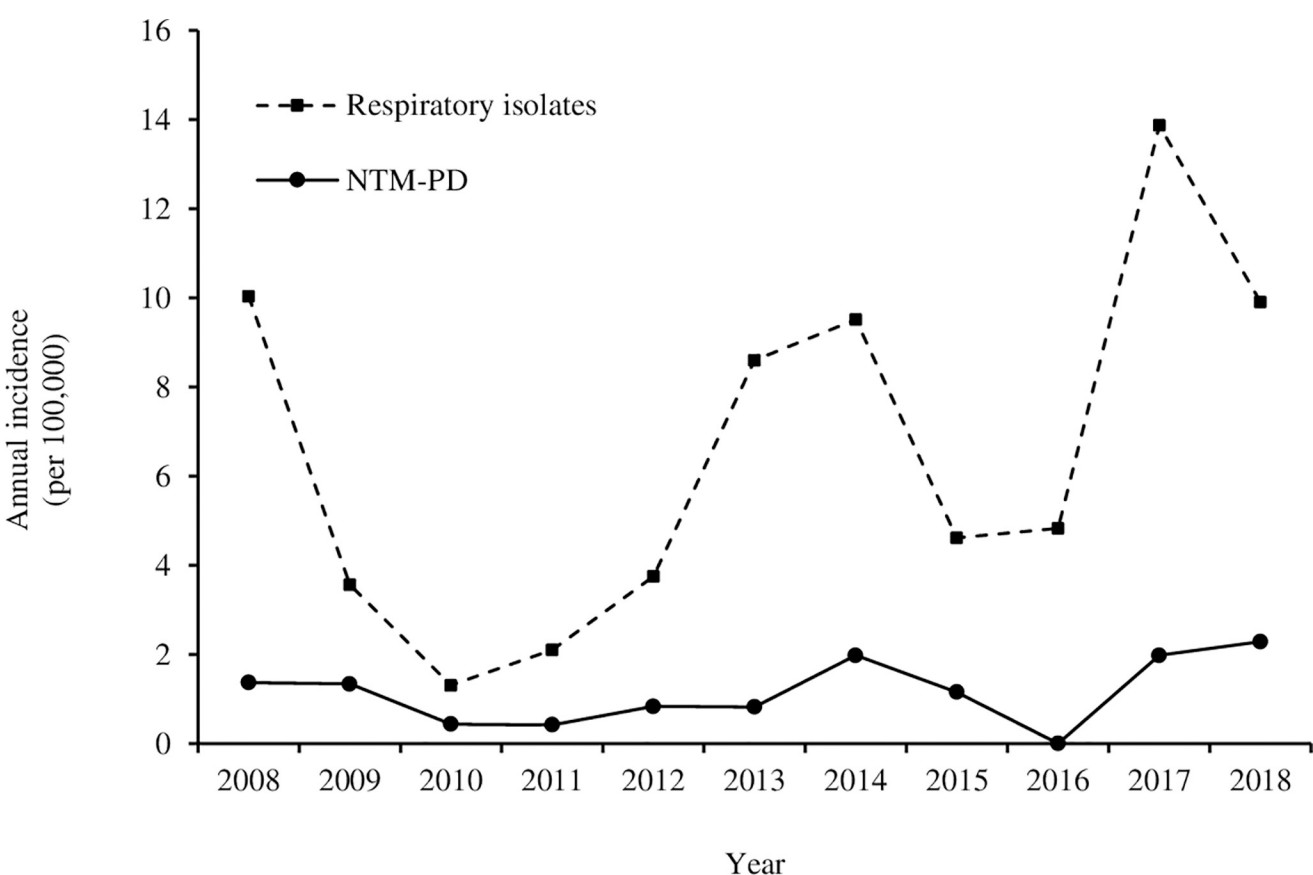

*PNTM: Pulmonary Non-Tuberculous Mycobacteria*

*NTM-PD: Non-Tuberculous Mycobacteria Pulmonary Disease*

**Fig 2. Incidence of PNTM infections in French Guiana (2008–2018).**

The average annual incidence over the study period was 6.17 / 100,000 inhabitants / year for overall PNTM positive respiratory sample and 1.07 / 100,000 inhabitants / year for NTM-PD. The incidence in French Guiana is lower compared to mainland France incidence, which showed a range between 0.72 to 0.74 / 100,000 inhabitants / year for NTM-PD between 2001 and 2003, followed by 1.3 to 13.6 / 100,000 inhabitants in 2016. It is also important to mention that the range of the incidence in mainland France was dependent on the region that was studied [12,13]. The differences in the incidences could be explained by different study periods, inclusion criteria and NTM-PD definitions. The current study used the 2007 ATS/IDSA criteria. All patients were discussed and classified into either NTM-PD or casual NTM isolation group, which could perhaps underestimate our NTM-PD incidence. However, similar incidence for respiratory specimen and NTM-PD was found in Brazil in 2005 (5.3 and 1.0 / 100,000 inhabitants / year respectively) and Canada (6.5 / 100,000 inhabitants / year on average between 1990 and 2006) also with an increasing in incidence over time [14].

**Table 2. Identified PNTM strains.**

| PNTM species | NTM-PD n = 31 (17%) | Casual PNTM isolation n = 147 (83%) | Overall N = 178 (100%) |
|---|---|---|---|
| Slow Growing Mycobacteria (SGM): | | | |
| M. avium | 16 (52%) | 18 (12%) | 34 (19%) |
| M. intracellulare | 9 (29%) | 25 (17%) | 34 (19%) |
| M. asiaticum | 0 | 2 (1%) | 2 (1%) |
| M. interjectum | 0 | 3 (2%) | 3/ (2%) |
| M. kansasii | 0 | 3/ (2%) | 3 (2%) |
| M. lentiflavum | 0 | 1 (1%) | 1 (1%) |
| M. genavense | 1 (3%) | 0 | 1 (1%) |
| M. scrofulaceum | 0 | 7 (5%) | 7 (4%) |
| M. simiae | 0 | 1 (1%) | 1 (1%) |
| M. szulgai | 0 | 2 (1%) | 2 (1%) |
| M. xenopi | 0 | 1 (1%) | 1 (1%) |
| M. gordonae | 0 | 7 (5%) | 7 (4%) |
| M. celatum | 0 | 2 (1%) | 2 (1%) |
| Rapid Growing Mycobacteria (RGM): | | | |
| M. abscessus | 5 (16%) | 5 (3%) | 10 (6%) |
| M. fortuitum | 0 | 34 (23%) | 34 (19%) |
| M. smegmatis | 0 | 4 (3%) | 4 (2%) |
| M. mucogenicum | 0 | 1 (1%) | 1 (1%) |
| Unidentified | 0 | 31 (21%) | 31 (17%) |

Our findings also showed an increasing trend for NTM isolates from 2013 onwards. This finding could be reflective of the improved diagnostic methods, since the culture on liquid medium of *Mycobacteria* was only available from 2012 onwards at Pasteur Institute of French Guiana. Nevertheless, 17% of NTM were unidentified. From 2008 to 2015 the identification of NTM was performed at the Pasteur Institute of Guadeloupe and private laboratories using the GenoType Mycobacterium CM and AS kits. This technique unfortunately, does not allow the identification of all species of NTM. More acurate identification of isolates by 16S rDNA and *hsp65* sequencing, in future studies, would be helpful in discovering new species, similar to the finding in French Polynesia [15]. Some strains were sent to the Referential National Center (RNC) in Paris for antibiogram susceptibility.

**Table 3. Significance of main PNTM group in French Guiana.**

| PNTM species | NTM-PD n = 31 (17%) | Casual PNTM isolation n = 147 (83%) | Overall n = 178 | OR | 95% CI | p |
|---|---|---|---|---|---|---|
| M. avium complex | 25 (81) | 43 (29) | 68 (38) | 10, 08 | 3,65–31,75 | < 0,001 |
| Other SGM* | 1 (4) | 29 (28) | 30 (17) | 0,10 | 0,01–0,69 | 0,008 |
| M. abscessus complex | 5 (16) | 5 (3) | 10 (6) | 5,46 | 1,15–25,23 | 0,02 |
| M. fortuitum | 0 | 34 (23) | 34 (19) | 0 | 0–0,42 | <0,001 |
| Other RGM** | 0 | 5 (3) | 5 (3) | 0 | 0–12,51 | 0,55 |
| Unidentified | 0 | 31 (21) | 31 (17) | 0 | 0–0,47 | 0,01 |
| Non pathogenic $ | 0 | 17 (12) | 17 (10) | 0 | 0–0,80 | 0,03 |
| M. tuberculosis coinfection | 2 (7) | 5 (4) | 7 (4) | 1,97 | 0,17–12,74 | 0,35 |

* Slow Growing Mycobacteria

** Rapid Growing Mycobacteria

$ M.lentiflavum, M.scrofulaceum, M.simiae, M.gordonae, M.mucogenicum

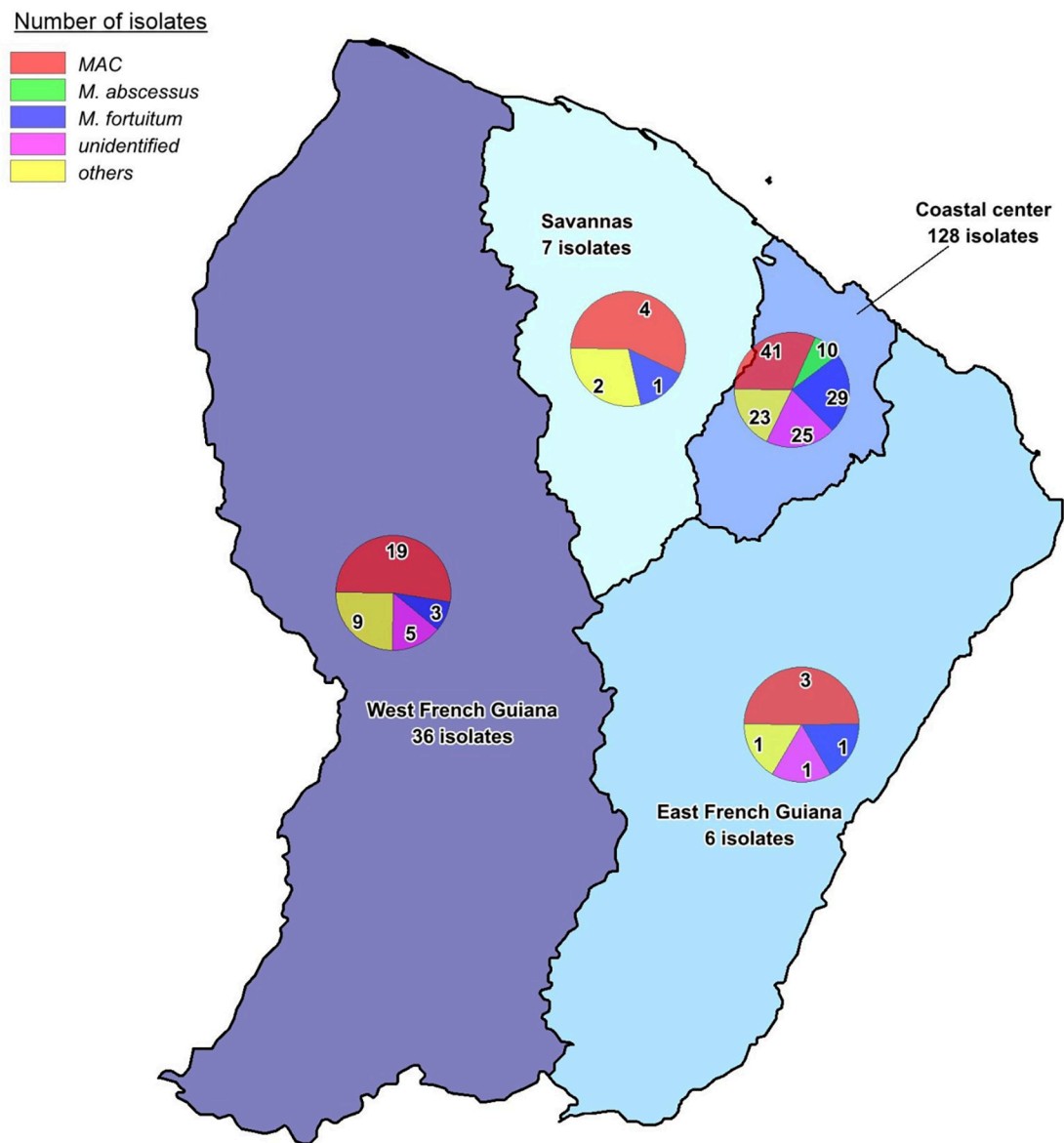

PNTM: Pulmonary Non-Tuberculous Mycobacteria
MAC: Mycobacterium Avium Complex

* Place of residence was unknown for one of the patients.
* Map was created using BdCarto NGI's (National Geographic Institute) map database
Base layer of the map:
https://geo.data.gouv.fr/fr/datasets/150be61acdce4635daaa8c30fe1f88f088f2e868
Layer created from IGN topo comics (2014)
The original data is under Open License 2.0, the description of which is:
https://www.etalab.gouv.fr/licence-ouverte-open-licence/

**Fig 3. Distribution of respiratory isolates of PNTM in French Guiana.** Map was created using BdCarto NGI's (National Geographic Institute) map database. Base layer of the map: https://geo.data.gouv.fr/fr/datasets/150be61acdce4635daaa8c30fe1f88f088f2e868. Layer created from IGN topo comics (2014). The original data is under Open License 2.0, the description of which is: https://www.etalab.gouv.fr/licence-ouverte-open-licence/.

This study showed an increasing trend in the number of NTM-PD from 2017 onwards. Other studies had shown an increasing incidence of NTM-PD between 2008 and 2015 [16]. NTM-PD incidence increases for elderly patients [17]. However, French Guiana's population is much younger compared to mainland France (median age 28.5 versus 40 in mainland France in 2015). Precariousness is also notably more important (unemployment rate 21.5% in 2015) along with a constantly increasing birth rate [18]. Among the studied population, 46% of patients had HIV infection. French Guiana remains the French territory with the highest tuberculosis declaration rate in 2017 reaching 32.5 / 100,000 inhabitants (2 times higher than in Ile de France and Mayotte Island in the Indian Ocean, 6 times more than in other French regions) [6,19]. These high number might be associated with a significant migratory flow and high HIV infection rate (number of HIV positive findings in 2015: 743 versus 89 per million people in mainland France).

Even in areas where tuberculosis is of major importance, infection with non-tuberculous mycobacteria needs to be considered [20]. Real incidence of NTM-PD may be underestimated, since NTM-PD is not a reportable disease, contrary to tuberculosis. Hence, the creation of a surveillance network of mycobacteria, would probably allow for more precise results.

Additionally, there is a possibility of underestimation of the results due to significant migratory flow in French Guiana, and the likely difficult access to health care [18].

In our study, the main species found in respiratory isolates were *MAC*, followed by *M. fortuitum* and *M. abscessus*. NTM-PD were due to *MAC*, *M. abscessus* and *M. genavense*. *MAC* is the first NTM found worldwide [21,22] and showed an even distribution across the different regions of French Guiana (Fig 3). Similar results were found in the study by Streit et al [10]. His and Cadelis et al's teams have shown similar mycobacterial epidemiology in the French West Indies (Guadeloupe and Martinique Islands)[23]. These are tropical French overseas territories, separating the Atlantic Ocean and the Caribbean Sea. The results could be explained by environmental similitaries: climate, atmospheric water vapor, soil, or water exposure. Nevertheless, it differs from the findings of a study in Reunion island, another French island in the Indian Ocean, where *M. simiae* was seen in 15% of cases of PNTM, while only one isolate was found in this study [24]. *M. simiae* seems to be more prevalent in contaminated water supplies. Underlying diseases such as cystic or non-cystic fibrosis bronchiectasis and diabetes mellitus were commonly described in others studies, however, they were not frequent in this study [10,24,25].

Regional differences support the hypothesis that habitus and environment are involved in the epidemiology of NTM. In French Guiana, *M. abscessus* was isolated in patients only in the coastal center region whereas in mainland France the proportion of *M. xenopi* was more important in Paris [12].

Respiratory mycobacteriosis in French Guiana in this study were due to *MAC*, *M. abscessus* and *M. genavense*. Contrary to mainland France where *M. xenopi* and *M. kansasii* were the second and third species to have a clinical impact after *MAC* [12,26].

*M. xenopi* and *M. kansasii* are more prevalent in Chronic Obstructive Pulmonary Disease (COPD) which concerned only 33% of patients of this study [27,28].

*M. xenopi* is rarely found in South America compared to *M. kansasii* due to the continental and regional differences [22,29]. *M. kansasii* seems to be prevalent in industrial regions and is also associated with lifestyle of the patients. For e.g. same species and similar distributions these species can be seen in Brazil and French Guiana. However, a great variability in the distribution of *M. kansasii* was seen. This variability in distribution was seen due to regional differences (presence of heavy and mainly mining industries) and lifestyle (swimming pools, whirlpool baths, hammam etc) [30–32]. In Argentina, the most important pathogen was *MAC* particularly because of HIV co-infected patients [33]. In Colombia, *MAC* and *M. abscessus*

were predominant as well [34]. In the current study, 16% of patients of this study had a history of tuberculosis and 46% had HIV infection.

Atypical mycobacteria differences in isolates seem to be correlated with water vapor exposure, which is a more important factor in tropical territories compared to Europe. [35,36]. Water vapour exposure was reported in studies in the USA, where NTM-PD were more predominant on the South coast and Hawaii [37,38]. As suggested by Prevots et al, the use of municipal water could explain why PNTM were more often isolated on the coastal central area, as the main city of Cayenne is located here in French Guiana, where there is significant use of municipal water [36]. French Guiana's soil is rich in manganese, clay, and the coastal swampy soils have an acidic pH, as described by Joseph Falkinham [39,40]. The temperature of the sea water in French Guiana is quite stable, generally around 27–28˚C, while the water in the rivers and creeks is a little cooler. Because of the high temperature throughout the year, water heaters are used less compared to mainland France. In coastal towns and villages, residential areas are well equipped with running water. However, in the slums of the main towns, rainwater is often the primary water source. In the remote communities far from the coast, rainwater collection tanks and small boreholes are used for the daily water supply [41].

*M. abscessus* is the second or third mycobacteria responsible for diseases on the American continent [29]. It is often associated to underlying pulmonary diseases such as bronchiectasis or COPD [42,43]. In French Guiana, while *M. abscessus* was the second specie to have a clinical impact, however bronchiectasis and COPD was seen in only 5 and 17% (respectively) of our population. Rapid Growing Mycobacteria (RGM) were predominant in the coastal center region. *M. abscessus* was found only in the coastal central region. Adjemian and al showed a highest prevalence of *M. abscessus* around the tropical climate [44]. Because of antibiotic resistance, the pulmonary disease caused by *M. abscessus* remains difficult to treat. As previously described, *M. fortuitum* was found in 19% of patients, and an associated with pulmonary disease was not seen [21].

There are several limitations to this study. Since this is a retrospective study, missing data is an unfortunate but an inherent limitation of the study design. As prevalences could not be assessed due to the missing data, we thereby focused on incidences. The incidences were calculated on an average annual incidence. Another limitation could be the lack of compulsory notification of NTM samples to public instances which would adversely affect the reporting of the number of cases. Nevertheless, the collection of respiratory samples was done exhaustively, wherein we included every laboratory analyzing these bacteria. Secondly, every chest scan was re-read by a pulmonologist but not systematically by a radiologist during the study, which could have lead to a classification bias. Thirdly, we used the ATS/IDSA 2007 because the ATS/IDSA 2020 criteria were published after the study had been completed. Nevertheless, 50% of NTM-PD patients had a positive protected sample in addition to their expectorated sputum samples and could agree with the 2020 ATS/ERS/IDSA/ESCMID criteria as well.

## Conclusion

Although the incidence of tuberculosis remains the highest in French Guiana, NTM should be given due consideration since its incidence is comparable to other countries and metropolitan France. The microbial epidemiology of NTM in French Guiana is different in comparison to the mainland France due to geographical, environmental and clinical differences, however it is similar to South America and the Caribbean territories. *MAC*, *M. fortuitum* and *M. abscessus* are the most frequently found species. *MAC* and *M. abscessus* are mainly responsible for diseases linked to NTM-PD and could be responsible for the adverse social and economic consequences due to the various difficulties associated with treating these diseases in a comparatively more precarious and immunosuppressed population than in mainland France.

## Acknowledgments

The autor thanks Mathieu Nacher (Inserm 1424, French Guiana), Hugo Testaert (Pneumology Department, University Hospital of Guadeloupe) and Sadia Khan (Bordeaux University, INSERM, Bordeaux Population Health Research Center, team: EPICENE, UMR1219, Bordeaux, France) for correcting the English. The autor thanks Gilles Chaptal for the realization of the map of French Guiana. The autor also thanks the French Guiana PNTM working group: Philippe Abboud, Houari Aissaoui, Alain Berlioz-Arthaud, Bastien Bidaud, Denis Blanchet, Anne-Marie Bourbigot, Mathilde Boutrou, Jean-Michel Cauvin, Fabrice Flament, Claire Grenier, Hatem Kallel, Anthony Le Labourier, Dominique Louvel, Aude Lucarelli, Aba Mahamat, Alessia Melzani, Balthazar N'tab, Richard Naldjinan-Kodbaye, Milko Sobeski, Antoine Talarmin, Stéphanie Thomas, Valentine Travers, Vincent Vantilke, Tania Vaz, Guillaume Vesin, Gaëlle Walter.

## Author Contributions

**Conceptualization:** Milène Chaptal, Loïc Epelboin.

**Formal analysis:** Emmanuel Beillard.

**Methodology:** Timothée Bonifay.

**Resources:** Geneviève Guillot, Veronique Jacomo.

**Validation:** Milène Chaptal.

**Writing – review & editing:** Claire Andrejak, Stéphanie Guyomard-Rabenirina, Magalie Demar, Sabine Trombert-Paolantoni, Emilie Mosnier, Nicolas Veziris, Felix Djossou.

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
