## [Decision Letter · Decision Letter 0]

30 May 2021

Dear Dr. Chaptal,

Thank you very much for submitting your manuscript "Epidemiology of infection by pulmonary non-tuberculous mycobacteria in French Guiana 2008-2018." for consideration at PLOS Neglected Tropical Diseases. As with all papers reviewed by the journal, your manuscript was reviewed by members of the editorial board and by several independent reviewers. In light of the reviews (below this email), we would like to invite the resubmission of a significantly-revised version that takes into account the reviewers' comments. 

We cannot make any decision about publication until we have seen the revised manuscript and your response to the reviewers' comments. Your revised manuscript is also likely to be sent to reviewers for further evaluation.

Sincerely,

Joseph M. Vinetz

Deputy Editor

Joseph Vinetz

Deputy Editor

Reviewer's Responses to Questions

**Key Review Criteria Required for Acceptance?**

**Methods**

-Are the objectives of the study clearly articulated with a clear testable hypothesis stated?

-Is the study design appropriate to address the stated objectives?

-Is the population clearly described and appropriate for the hypothesis being tested?

-Is the sample size sufficient to ensure adequate power to address the hypothesis being tested?

-Were correct statistical analysis used to support conclusions?

-Are there concerns about ethical or regulatory requirements being met?

Reviewer #1: the analysis is simple and does not require additional work

Reviewer #2: The objectives of the study are clearly defined and presented. The authors used an appropriate study design to reach the objectives of the study, the population that was used was adequately described and the sample size, while quite small, is appropriate for such a rare phenotype. The authors made use of appropriate statistical models and all ethical considerations for the study has been met.

Reviewer #3: The methods were a retrospective review of non-tuberculous mycobacterial disease (specifically pulmonary in origin) in French Guiana in toto. Three major hospitals participated in the inquiry. This was done by first doing an inquiry of mycobacterial cultures from 2008-2018; once patients were identified they went through a second inquiry to look at radiographic evidence of disease and the plurality of the cultures (and of course their source). 

The study population was appropriately studied based on the specific aims of the authors and based on the organisms in question the population studied would be limited in number. 

Statistic analysis was done via STATA, and demographic collected via Microsoft Excel; this kind of analysis is on par with other studies of the same caliber. 

The authors state that this their studies were notified to the "National Commission for Data Protection", but it is unknown if this study was approved by a governing board (such as an institutional review board or IRB) for human protections or if there was any study protocols which took safeguards into account for this project. There is no indication that there are ethical issues, but this should be explained in the Methods section of the manuscript.

**Results**

-Does the analysis presented match the analysis plan?

-Are the results clearly and completely presented?

-Are the figures (Tables, Images) of sufficient quality for clarity?

Reviewer #1: the results are sufficiently clear given the simplicity of the analysis

Reviewer #2: The results as presented does match the analysis plan. The results are clearly presented and are complete, while all figures and tables are of sufficient quality. There are some editorial issues with the figures and tables that I will point out under the Editorial and Data Presentation Modification section of this review.

Reviewer #3: The results were interesting and they are clearly presented. This is especially true since they mirror what is seen in other tropical/subtropical areas of the planet that have NTM infections. 

Tables were adequate, however there should be legends for both the tables and figures. For the figures: Figure 1 appeared to be of poor resolution and thus was blurry to read; Figure 2 had line markers that were difficult to see, and thus using a different scheme for the appearance of the two lines would be better; Figure 3 was satisfactory, yet the font color for the numbers within the pie charts were difficult to read against the background and thus should be more uniform with a darker color.

**Conclusions**

-Are the conclusions supported by the data presented?

-Are the limitations of analysis clearly described?

-Do the authors discuss how these data can be helpful to advance our understanding of the topic under study?

-Is public health relevance addressed?

Reviewer #1: (No Response)

Reviewer #2: The conclusions that the authors draw for their study are supported by the data. They do acknowledge that there are some limitations to the study and these study limitations are clearly and frankly discussed. The impact on public health is not discussed in detail and the authors do not really discuss how the results of their study advances what is known about NTMs. However, the study is the first to investigate and report on the prevalence of NTMs in French Guiana, it does have significant public health issues for the region.

Reviewer #3: The final conclusions are supported by the data presented and the limitations were appropriate for this kind of study. The real issue with this section was the Discussion. It was not presented in a way that the reader could easily follow. Also, the vernacular seemed to be less put together contrasted with what was seen in the rest of the manuscript. This made the issue of understanding the conclusions within the discussion more problematic and thus less impactful. A good example of this is noted on Pages 8-10. The authors go from documenting Canadian rates of NTM prevalence and then diving into France and what is seen there to coastal Brazil and then back to France when it comes to their MTB prevalence. Also there is no data discussed about what is transpiring in other nations such as the United States, Mexico or other countries of South America. Furthermore, there are minimal mentioning of association with other diseases which are more frequently associated with NTM mentioned in the manuscript (as was illustrated in Table 1). One would ask the authors if there was any data to suggest that M. xenopi is seen in more temperate/cooler climates compared to M. abscessus which is seen in more tropical/subtropical climates? Data from Rebecca Prevots group at National Institutes of Health (USA) and Theodore Marras' group at University of Toronto have both looked at this in the past. Both are in the references, but their data is important to review and thus make the argument for the rigor of what is documented in these data. 

Finally, would include any supporting conclusions about whether soil composition, potable water methods used in French Guiana or other factors could be supporting the results seen in this investigation. This is touched upon, but there is no real hypothesis and literature review regarding this topic. Manuscripts by Joseph Falkinham's group has documented this well and has reasoning for potential hypotheses that would make the case stronger for these data.

**Editorial and Data Presentation Modifications?**

Reviewer #1: (No Response)

Reviewer #2: There are a few issues with the way that the results are presented.

1) In table 1, the 1.1.1.1.1.1. in the CCI score (in column 2) should be removed.

2) In table 1, the ** denoting cerebrovascular disease mentions 6 stroke, 1 cerebral amyloidosis and 1 mental retardation. This adds up to 8, but in the table, it indicates that there are 9 cerebrovascular disease. Please reconcile what is in the legend with what is in the table. 

3) There needs to be consistency between the labeling and heading and titles used in the table. In table 2, "Pulmonary NTM" is used while in table 3, "PNTM" is used. Please choose one and stick with it.

4) In the tables, the number of patients in each category (n=31 for NTM-PD and 147 for Causal PNTM) is already stated in the heading, so there is no need to have the number in each cell as well. For example, instead of writing 16/31 (52%), it is sufficient to just write 16 (52%).

5) Table 3 shows a comparison between the NTM-PD and Causal PNTM, however this is not very clear from the manuscript. Please make it clear in the text that this what is shown in table 3. Also, this is should be discussed more in the results.

6) In table 3, some of the p-values are in bold. I assume that these indicate significant differences between the two groups. However, there are some p-values which are less than 0.05 are not shown in bold. This needs to be fixed or changed in the table.

7) Figure 3: The pie charts in the chart key are confusing and not needed. These seem to denote the number of isolates, which are indicated as 20, 10 and 2. This does not correspond to the pie charts on the figure. Why are some of the pie charts on the figure bigger than the others?

8) Figure 3: The total number of isolated indicated on the table is 177, while in the tables the total number of isolates in the study is indicated as 178. Please check and fix. I noticed that in figure 3, the total number of MAC is 67, while in the tables it is 68.

Reviewer #3: As mentioned above, there needs to be a better approach to the figures/tables as they are the fastest way to convey data to the readers of PLOS NTD. In addition, the discussion needs to be more streamlined in it's message and use appropriate English.

**Summary and General Comments**

Reviewer #1: The article is an interesting addition to the literature. however, there is room for improvement

the introduction should explicitly mention that the particular amazonian ecosystem makes the question interesting at least in my view this is what the added value of the paper is

in line with this i find the discussion of the differences between locations canada france etc hard to follow. it may be helped by a comparative table of the ranking of different studies in different sites (tropical non tropical) proportion of hiv perhaps

the authors mention that "This could be one of the explanations of the

low proportion of M. kansasii found in French Guiana since this species is the adage of HIVpositive

patients with CD4> 200 / mm3 (18). " i think this is wrong because most hiv patients in french guiana have cd4 counts above 200

although it is a minor point the english language should be improved.

Reviewer #2: In general the article is well-written with and easy to understand. There are concerns about the figures and tables that I highlighted. Some additional comments:

1) Please make sure that all abbreviations are adequately defined. For example, on page 7, the abbreviation INSEE is used. Please define this abbreviation the first time it is used.

2)On page 10, the authors talk about a decrease in the number of HIV positive cases and says that it remains high with 743 million inhabitants (versus 89 million in mainland France). Do the authors mean that there were 743 million people wit HIV in French Guiana vs 89 Million in mainland France? This cannot be possible. Please rephrase the sentence so that it explains these numbers.

Reviewer #3: I have been asked to review Chaptal et al. "Epidemiology of infection by pulmonary non-tuberculous mycobacteria in French

Guiana 2008-2018." This is a retrospective review of cases of non-tuberculous mycoabcteria (NTM) in the French territory of Guiana in South America. This was done in collaboration with academic institutions in both the territory and in France as well as with a consortium of NTM researchers across the Atlantic. Since this is an emerging area of not only infectious diseases, but also in epidemiology, this manuscript has an important impact message to the readers of the journal. Thus, the significance of the study is not only appropriate, but also is necessary as mycobacteria are ubiquitous in the environment as well as becoming studied more frequently as a cause of human disease. The study methods and execution were sound. The real weaknesses were not with the study, but how the data was put together and illustrated. This was evident in the Figures/Tables and the Discussion. 

Thus, the manuscript requires major revision of the following:

1) addition of an ethical statement regarding the use of patient sensitive information and how that information was reviewed and safeguarded. This is generally important when dealing with data mined from patient charts and thus patient information should not be compromised. 

2) Figures and tables need to be more clear and the data illustrated in a more appropriate form. All figures/tables should come with legends explaining the data. 

3) The Discussion section appeared to not be written in the same format as the rest of the manuscript and thus needs major revision and rewording to accomplish the illustration of what these data demonstrate. Someone proficient in the scientific use of English would be helpful to have on hand to proofread the manuscript. Lastly, the data here are important to document, however the hypotheses as to why these data are the way the way they are is critical to what the authors are trying to accomplish. A more intense literature review and understanding of what is happening in environment and making the correlation with the clinical human disease is paramount. 

What is documented here is also documented in the above sections.

PLOS authors have the option to publish the peer review history of their article (what does this mean?). If published, this will include your full peer review and any attached files.

Reviewer #1: No

Reviewer #2: No

Reviewer #3: No
---

## [Decision Letter · Decision Letter 1]

1 Nov 2021

Dear Dr. Chaptal,

Thank you very much for submitting your manuscript "Epidemiology of infection by pulmonary non-tuberculous mycobacteria in French Guiana 2008-2018." for consideration at PLOS Neglected Tropical Diseases. As with all papers reviewed by the journal, your manuscript was reviewed by members of the editorial board and by several independent reviewers. The reviewers appreciated the attention to an important topic. Based on the reviews, we are likely to accept this manuscript for publication, providing that you modify the manuscript according to the review recommendations. 

Sincerely,

Joseph M. Vinetz

Deputy Editor

Editorial comments: The editors concur with the review. Considerable effort needs to be made to improve the writing, as this journal does not provide copyediting services. The article is published as is from the authors.

Reviewer's Responses to Questions

**Key Review Criteria Required for Acceptance?**

**Methods**

-Are the objectives of the study clearly articulated with a clear testable hypothesis stated?

-Is the study design appropriate to address the stated objectives?

-Is the population clearly described and appropriate for the hypothesis being tested?

-Is the sample size sufficient to ensure adequate power to address the hypothesis being tested?

-Were correct statistical analysis used to support conclusions?

-Are there concerns about ethical or regulatory requirements being met?

Reviewer #3: The methods of this revised manuscript were not really modified, but requested changes were made to the points from the reviewers. This includes a statement about the institutional review board for these types of studies that was requested by this reviewer. As mentioned previously, the study design does address the stated objectives and discusses the methods used in this epidemiological analysis.

**Results**

-Does the analysis presented match the analysis plan?

-Are the results clearly and completely presented?

-Are the figures (Tables, Images) of sufficient quality for clarity?

Reviewer #3: The revisions made by authors as suggested by the reviewers were done and more. The results go along with the aims and hypotheses made by the authors at the beginning. The figures still are of low quality for PLOS NTD and would strongly advise the authors to counsel with a professional at making figures for journals for a revision. These figures are what readers of PLOS NTD are going to be looking at while reading the prose of the manuscript. Quality control for journals can only do so much when the figures are out of focus based on the medium in which they were created.

**Conclusions**

-Are the conclusions supported by the data presented?

-Are the limitations of analysis clearly described?

-Do the authors discuss how these data can be helpful to advance our understanding of the topic under study?

-Is public health relevance addressed?

Reviewer #3: Yes, the conclusions are supported by the data analysis and confirm the authors suspicions based on the climate and the socioeconomic status of the investigation subjects. The comparisons to both it's southern neighbor and mainland France do offer some perspective on this. It would be interesting to also note some of the basic information about water temperature in French Guiana and if water heaters are used as compared to mainland France. This is the hypothesis of Dr. Marras and colleagues at U Toronto. Some basic information on this as well as how much of the country uses well water or rain catchment for their fresh water source would be helpful in solidifying the conclusions based on their hypotheses.

**Editorial and Data Presentation Modifications?**

Reviewer #3: Please see above in the 'Conclusions' section of the review. English used in the manuscript is better, yet would still suggest that this is reviewed by one who has published in an English-language journal to optimize the vernacular and enhance the point they are trying to make with this manuscript.

**Summary and General Comments**

Reviewer #3: This is the second review for the manuscript entitled, "Epidemiology of infection by pulmonary non-tuberculous mycobacteria in French Guiana 2008-2018." The manuscript was reviewed previously by this reviewer; the premise of the study is to determine the epidemiology of non-tuberculous mycobacterial disease in French Guiana, which is part of France. The aim and hypotheses are the same as was the data analysis, which demonstrates that Mycobacterium avium Complex and Mycobacterium abscessus dominate as opposed to mainland France which has more M xenopi as part of the infections of NTM there. This is an important study as it demonstrates data from a part of South America which has minimal reporting. 

The manuscript still has some lingering issues; those being manuscript English, details about fresh water source for the populations in the various municipalities and Figures, which need to be improved dramatically for PLOS publication standards. 

Minor revision should be instituted for the above commentary prior to acceptance.

PLOS authors have the option to publish the peer review history of their article (what does this mean?). If published, this will include your full peer review and any attached files.

Reviewer #3: No

Figure Files:

Data Requirements:

Reproducibility:

References

---

## [Editor Report · Decision Letter 2]

4 Jun 2022

Dear Dr. Chaptal,

Thank you very much for submitting your manuscript "Epidemiology of infection by pulmonary non-tuberculous mycobacteria in French Guiana 2008-2018." for consideration at PLOS Neglected Tropical Diseases. As with all papers reviewed by the journal, your manuscript was reviewed by members of the editorial board and by several independent reviewers. The reviewers appreciated the attention to an important topic. Based on the reviews, we are likely to accept this manuscript for publication, providing that you modify the manuscript according to the review recommendations. 

Sincerely,

Joseph M. Vinetz

Deputy Editor

Joseph Vinetz

Deputy Editor

Figure Files:

Data Requirements:

Reproducibility:

References

---

## [Editor Report · Decision Letter 3]

22 Jul 2022

Dear Dr. Chaptal,

We are pleased to inform you that your manuscript 'Epidemiology of infection by pulmonary non-tuberculous mycobacteria in French Guiana 2008-2018.' has been provisionally accepted for publication in PLOS Neglected Tropical Diseases.

Best regards,

Joseph M. Vinetz

Section Editor

Joseph Vinetz

Section Editor

---

## [Editor Report · Acceptance letter]

1 Sep 2022

Dear Dr. Chaptal,

We are delighted to inform you that your manuscript, "Epidemiology of infection by pulmonary non-tuberculous mycobacteria in French Guiana 2008-2018.," has been formally accepted for publication in PLOS Neglected Tropical Diseases.

Best regards,

Shaden Kamhawi

co-Editor-in-Chief

Paul Brindley

co-Editor-in-Chief
